# Hash Layers For Large Sparse Models

**Stephen Roller**  **Sainbayar Sukhbaatar**  **Arthur Szlam**  **Jason Weston**

Facebook AI Research

## Abstract

We investigate the training of sparse layers that use different parameters for different inputs based on hashing in large Transformer models. Specifically, we modify the feedforward layer to hash to different sets of weights depending on the current token, over all tokens in the sequence. We show that this procedure either outperforms or is competitive with learning-to-route mixture-of-expert methods such as Switch Transformers and BASE Layers, while requiring no routing parameters or extra terms in the objective function such as a load balancing loss, and no sophisticated assignment algorithm. We study the performance of different hashing techniques, hash sizes and input features, and show that balanced and random hashes focused on the most local features work best, compared to either learning clusters or using longer-range context. We show our approach works well both on large language modeling and dialogue tasks, and on downstream fine-tuning tasks.

## 1 Introduction

Recent studies of Transformer models have shown a clear trend towards improvements with scale in data and model size [1], mirroring the same trend in Machine Learning more generally. However, when architected naively, larger (in terms of parameter count) models are slower to train and to evaluate; and at extreme scale, with current computer systems, necessitate complex engineering to facilitate communication between workers. To address these challenges, researchers have studied Mixtures-of-Experts (MoE) models [2, 3, 4, 5, 6, 7, 8], where a "gater" routes computation through a sparse subset of the weights of the model (the "expert modules"). Specifically in the setting of Transformers for Natural Language Processing (NLP), recent approaches have led to state of the art performance in language modeling [8]. MoE models allow increasing the number of parameters in the model while holding steady the number of computations that affect a given sample.

A key component to a MoE model is the routing (gating) strategy. While MoE models can be computationally advantageous per parameter compared to a dense model, they might be functionally less powerful per parameter. A poor routing strategy might lead to expert modules that are not properly specialized (essentially making a stochastic ensemble model); or overly specialized, using the data assignment function to overfit. Meanwhile, the routing strategy itself must be efficient.

A standard approach is to train a layer of weights that makes the routing decision based upon the input to the layer to be routed. Classically, this may have been implemented with a softmax over the choice of expert modules, and fitted via backpropagation. However, a dense softmax requires all expert modules to run on all data points at train time, which negates the computational savings. Several works have shown that sparsity can be maintained during training, e.g. [9, 7, 8, 10]. In particular, Switch Transformers [8] select the top expert per token using a softmax over the token's hidden state, but require a load balancing term in the objective function or they can become imbalanced or degenerate, giving poor results. BASE Layers [10] employ a linear assignment algorithm to try to resolve the same problem.

35th Conference on Neural Information Processing Systems (NeurIPS 2021).

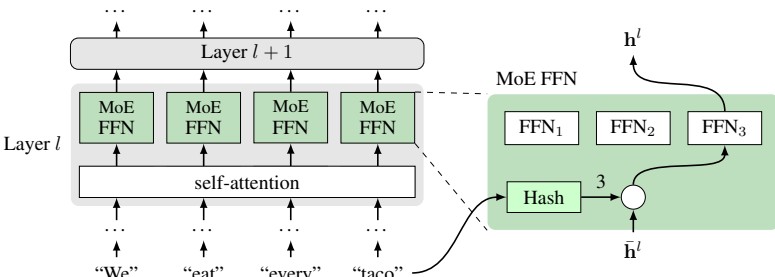

Figure 1: **Overview of the Hash Layer.** Tokens are routed to fixed expert modules based on their hash.

In this work, we describe a simple, sparse, efficient routing strategy based on hashing input tokens that is effective in the Transformers-for-NLP setting. We show this approach is effective on a number of datasets, comparing favorably to both Switch Transformers and BASE Layers. As the routing strategy requires no extra parameters, no change to the objective function or assignment algorithm, its simplicity means it is robust, fast and easy to implement. We provide detailed analysis to explain why our method works, and in which conditions. Given that when training very large models one may typically have only one shot given the required compute budget, and experimenters will be unable to try many parameter choices, we hence advocate our approach as a strong candidate for such a setting.

## 2 Background

Let us first introduce the Mixture-of-Experts setting where we apply our hash-based routing strategy. We use the same setting as [11, 8, 10] where a feedforward network (FFN) in a Transformer is replaced by its MoE version. Given a tokenized input sequence $\{x_1, x_2, \ldots, x_T\}$ of $T$ tokens, a representation for each token is computed in parallel by a standard Transformer [12]

$$\mathbf{h}_1^L, \mathbf{h}_2^L, \ldots, \mathbf{h}_T^L = \text{TRANSFORMER}(x_1, x_2, \ldots, x_T). \tag{1}$$

The Transformer consists of $L$ layers that computes final hidden states for each token, and each layer is composed of self-attention and FFN sublayers, where FFNs are two-layer fully connected networks

$$\bar{\mathbf{h}}_t^l = \text{SelfAttn}(\mathbf{h}_t^{l-1}) \qquad \mathbf{h}_t^l = \text{FFN}(\bar{\mathbf{h}}_t^l). \tag{2}$$

Here we omit skip-connections and normalization for brevity. We can then replace one or more of the FFN sublayers with expert modules. Replacing the FNN at layer $l$ with $K$ expert FFNs, their output is then mixed with some gating function $g(\cdot)$:

$$\mathbf{h}_t^l = \text{FFN}(\bar{\mathbf{h}}_t^l) \quad \rightarrow \quad \mathbf{h}_t^l = \sum_{i=1}^{K} g_i(\bar{\mathbf{h}}_t^l) \, \text{FFN}_i(\bar{\mathbf{h}}_t^l), \quad t = 1, \ldots, T, \tag{3}$$

where importantly each token is routed to a different mixture of experts, as the gating function depends on the token's specific hidden state $\bar{\mathbf{h}}_t^l$.

Sparse MoE methods assume gating values $g_i$ are often zero, so only a few experts need to be computed for better efficiency. As expert FFNs do not share parameters, the number of parameters increases with $K$ while the amount of computations per input token stays the same if the MoE FFN only routes to a single expert, and computation of $g_i$ is cheap. While this allows training of large capacity models with small compute budget, optimizing $g_i$ in the sparse setting can be tricky.

## 3 Method

In this paper we propose a simple gating mechanism that is especially efficient because only one expert is active, and it has no routing network parameters to be learnt. Recent work [11, 8, 10] has to learn parameters that determine the routing to expert modules based on hidden states, which have to be optimized in tandem with the expert weights themselves. This can potentially cause difficulty because during training membership for each expert is changing while it is trying to learn the mapping

for those members. We instead advocate for a fixed mapping to experts. Namely, by *hashing* the tokens into a fixed number of buckets, each bucket corresponding to an expert:

$$\mathbf{h}_t^l = \text{FFN}_{\text{hash}(x_t)}(\bar{\mathbf{h}}_t^l), \quad t = 1, \ldots, T. \tag{4}$$

While the FFN still takes the hidden state $\bar{\mathbf{h}}_t^l$ as input, our routing function uses the original input token $x_t$ rather than the hidden state, see Figure 1 for a graphical depiction. We are free to choose from various possible hash functions, which we will consider below. However, for training purposes, the hash function is fixed in advance, and in this way, our routing mechanism requires no training and has no adjustable parameters.

## 3.1 Hash Functions

Hash functions have long been employed throughout Computer Science [13], and can take a variety of forms. In our work, we generally employ pre-computed hash functions, which use a lookup table during learning – precomputed in advance – to map tokens to expert modules.

We consider several kinds of hash functions as possible choices for routing tokens to expert modules. The simplest is **Random Hash**, wherein we assign every token to a fixed, random expert at initialization. Due to the Zipfian distribution of token frequency, this naturally produces imbalance across the different expert modules. As balancing has been previously shown to be important for training MoE models [8, 10], we also consider **Balanced assignment**. In this method, we build the lookup table before training the model using the training data distribution by greedily assigning the most frequent tokens to the emptiest buckets. The resulting assignment structure is significantly more balanced than Random Hashing, but not perfect, as the frequency of some tokens exceeds the ideal distribution.

Random and Balanced hashing exploit the inductive bias of auto-regressive models and hash on the input token, but we also consider other possibilities: **Bigram Hash** uses the current and previous token $(x_{t-1}, x_t)$ rather than only the current token, while **Previous Token Hash** uses the previous token $x_{t-1}$, ignoring the current input. We also consider a sanity check which hashes based on the **Position** in the sequence, which we expect to have little impact, as absolute positions carry little information in natural language. Each of these hash functions is used to assess the value of the information being routed-on in our subsequent experimental analysis.

As an upper baseline, we also evaluate using an **Oracle Future Hash**, which hashes based on the *output token* $x_{t+1}$, rather than input token. This Oracle Hash checks how powerful routing decisions can be in solving a task. Similarly, we also consider **Predicted Future Token Hash**, which utilizes a baseline Transformer to make a prediction of the output token, and then hashes over this prediction.

**Clustered Hashes**    Based on the intuition that similar tokens may want to be routed to the same expert, we also experiment with **Clustered Hashes**. We obtain clusters by performing k-means clustering with a fixed number of clusters using token embeddings from a baseline Transformer model. Each expert is assigned a centroid, and tokens are assigned to their closest cluster.

**Dispersed Hashes**    We also consider the opposite hypothesis: that similar-tokens should be placed in *different* buckets, where the assumption is that very similar tokens need fine distinctions which requires more model capacity (hence assigning to different experts). To do this, we use the same k-means clusters as before, but distribute all tokens within each cluster equally across all buckets.

## 3.2 MultiHash Layers

In the standard FFN MoE approach, all $K$ expert modules have independent parameters, but here we consider another option. It is known in the hashing literature that multiple hashes can provide better allocations in many contexts [14]. We consider such schemes in the context of sparse routing. Let us assume we are given $N$ different hashing functions, and for a given input token $x$ we compute these hashes, denoted as $k_m = \text{hash}_m(x), \ m = 1, \ldots, N$. Assuming the usual expert FFN is a function $B(\text{relu}(A(\mathbf{h})))$ where $A : \mathbb{R}^d \to \mathbb{R}^D$ and $B : \mathbb{R}^D \to \mathbb{R}^d$, we split the linear layers into $N$ segments, $A_m : \mathbb{R}^d \to \mathbb{R}^{D/N}$ and $B_m : \mathbb{R}^D \to \mathbb{R}^{d/N}$. Then we compute:

$$\mathbf{v} = \text{relu}([A_{k_1}(\mathbf{h}), \ldots, A_{k_N}(\mathbf{h})]) \qquad \text{FFN}_{\text{MH}}(\mathbf{h}) = [B_{k_1}(\mathbf{v}), \ldots, B_{k_N}(\mathbf{v})].$$

That is, use hashing to select the parameters we are going to use for each segment, and then concatenate them together. The advantage is that we are now no longer reliant on the quality of a single hash function, but have multiple chances to produce good quality partitions. This perhaps can also be seen as analogous to the multi-head attention process already used in Transformers.

# 4   Related Work

Sparse MoE models, where only a few expert modules are active for any input, in particular in the context of NLP, have been studied recently in [6, 11]. In these works, the gating is learned via backpropagation, perhaps with a regularizer to encourage load balancing across experts. [8] showed that models in [11] can be successfully trained with each input assigned to exactly one expert. Another such approach for Transformers, where the routing is learned via solving a linear assignment problem, is studied in [10]. [15] uses a different approach, where product keys enable nearest neighbor search to select parameters. More generally, using MoE to trade off compute time (at the cost of possible data fragmentation) has a long history, see e.g. [3, 7].

The approach in this work is different from all of these in that the assignments use no learning whatsoever, and instead make use of the inductive biases possible in the setting of natural language. In particular, we use the fact that $n$-grams are themselves decent language models [16]. Thus this work is related to previous work attempting to combine neural and $n$-gram language models [17, 18, 19, 20, 21, 22].

Our work is also related to feature hashing in linear models and kernel methods [23, 24], where word or n-gram features are hashed to provide a new lower dimensional feature space. [23] showed that when performing such feature hashing the interaction between random subspaces is negligible with high probability. [25] uses hashing to compress neural networks, rather than increase their parameters as we do here. Work on long-context Transformers has recently used hashing techniques to speed up access to long-range token history via sparse self-attention patterns, particularly in Routing Transformers [26] and the Reformer [27]. In contrast, our work uses hashing to access a large set of parameters via sparse routing, rather than sparse access to input features.

# 5   Experiments

## 5.1   Tasks

**Pushshift.io Reddit**    We use a variant of Reddit discussions, which has also been used in several existing studies, see e.g. [28, 29, 30, 31]. Following [32], we use a previously existing Reddit dataset extracted and obtained by a third party and made available on pushshift.io [33], training to generate a comment conditioned on the full thread leading up to the comment, spanning 1.5B training examples. We use the same BPE dictionary as [34], comprising of 8008 tokens.

**RoBERTa+cc100en Data**    We use the same data used to train BASE [10], which consists of approximately 100B tokens, combining corpora used in RoBERTa [35] with the English subset of the CC100 corpus [36]. The GPT2 dictionary, of size 51200, is used for tokenization. For our seq2seq experiments, we arrange this data splitting by sentence to predict the next turn. We consider it as the originally intended language modeling task in our experiments comparing with BASE [10].

**Wikitext-103**    Wikitext-103 is a smaller language modeling benchmark [37] consisting of a collection of Wikipedia articles of over 100 million tokens, and a fixed vocabulary size of 270K tokens is provided. We view this as a seq2seq task in our experiments, again splitting by sentence.

**Downstream BST tasks**    Finally, we use the Blended Skill Talk (BST) dialogue tasks used in [34] after pushshift.io Reddit pre-training to evaluate fine-tuning performance of dense vs. sparse models.

## 5.2   Experimental Setup

**Seq2Seq Setup**    The majority of our experiments are carried out in ParlAI[1] platform using an encoder-decoder Transformer framework. We first train several standard (dense) Transformers, with

---

[1]`http://parl.ai`

Table 1: **Comparison of Models on pushshift.io Reddit.** We show three sizes of dense Transformer compared to Switch Transformers and using Hash Layers with various numbers of modules and sparse layers, e.g. 5x16 means 5 sparse layers with 16 modules each. All Switch and Hash Layer modules are built to the same computational complexity as the 11 layer baseline Transformer, but have more parameters; the larger dense models have similar total parameters, but use more compute.

| Model | Configuration | Params | Valid PPL | Test PPL |
|---|---|---|---|---|
| Baseline Transformer | layers=11, $d$=1024, $D$=4096 | 222M | 24.90 | 24.96 |
| Wider Transformer (more compute) | layers=11, $d$=2048, $D$=6144 | 755M | 23.32 | 23.38 |
| Deeper Transformer (more compute) | layers=22, $d$=1536, $D$=4096 | 755M | 22.72 | 22.78 |
| Switch Transformer | layers=11,modules=1x64, load_bal=0.1 | 751M | 23.65 | 23.73 |
| Hash Layer | layers=11,modules=1x64 | 751M | 23.16 | 23.23 |
| Switch Transformer | layers=11,modules=1x128, load_bal=0.1 | 1.28B | 23.52 | 23.58 |
| Hash Layer | layers=11,modules=1x128 | 1.28B | 22.89 | 22.95 |
| Switch Transformer | layers=11,modules=5x16, load_bal=0.01 | 852M | 23.19 | 23.25 |
| Switch Transformer | layers=11,modules=5x16, load_bal=0.1 | 852M | 23.00 | 22.93 |
| Hash Layer | layers=11,modules=5x16 | 852M | 23.21 | 23.27 |

Table 2: **Comparison of Models on RoBERTa+cc100en Data.** We compare a dense transformer with the same parameters as our sparse models, except with 1 sparse layer with 64 modules (1x64).

| Model | Configuration | Params | Valid PPL |
|---|---|---|---|
| Baseline Transformer | layers=11, $d$=1024, $D$=4096 | 266M | 28.85 |
| Switch Transformer | layers=11, modules=1x64, load_bal=0.1 | 795M | 27.41 |
| Hash Layer | layers=11, modules=1x64 | 794M | 26.99 |

2 encoder layers and either 11 or 22 decoder layers, following the structure in [34] for training on pushshift.io Reddit. We refer to the one with 11 layers and embedding size of $d = 1024$ and FFN hidden layer size of $D = 4096$ as our Baseline Transformer. We also train a "Wider" model with $D = 6144$, and a "Deeper" model with 22 decoder layers, and $D = 4096$. The Baseline model has 222M parameters, and the "Wider" and "Deeper" are selected to both have 755M parameters each. These models are compared to the Hash Layer methods detailed in section 3 and to Switch Transformers of the same sizes and settings. The load balancing for Switch is optimized on the validation set. For both Hash and Switch we use the "Baseline" Transformer size detailed above as the architecture that we add sparse routing layers to by replacing one or more of the original dense layers. All experiments are run for 100k updates; a table of hyperparameters is provided in subsection B.1.

**BASE Comparison**  While most of our analysis takes place in the setup described above with models up to 1.28B parameters, to test our methods at scale on larger sparse models, we adopt the BASE Layer setup [10] and code base[2] instead where we compare 4.5B parameter Hash and BASE Layer models. This setting uses pure language models rather than the Seq2Seq setup above. We use the architecture, data (RoBERTa+cc100en), and hyperparameters directly from [10], using either a single sparse routing layer consisting of 3 stacked FFNs ($D = 8192$) on the middle layer of a 25 layer network, or 3 routing layers evenly spaced in the network. In order to compare with BASE directly, we keep all hyperparameters fixed and only change the routing method; we use a balanced assignment Hash Layer in this case. We trained until 40k steps had been reached. A table of hyperparameters is provided in subsection B.2.

## 5.3   Results and Analysis

### 5.3.1   Comparison between Hash, Switch and Dense models

**Hash vs. Switch routing on a single layer**  We first compare a Hash layer (with balanced hash) to a Switch layer, on an otherwise dense Transformer, where sparse routing is performed on layer 7 of the decoder. Both methods use 64 expert FFNs with 751M total parameters. Results on pushshift.io Reddit are given in Table 1 (rows 4 and 5) and on the RoBERTa+cc100en data in Table 2 (rows 2 and 3). We find Hash Layers outperforming Switch on both datasets by about 0.4-0.5 perplexity.

---

[2]Made available within Fairseq [38].

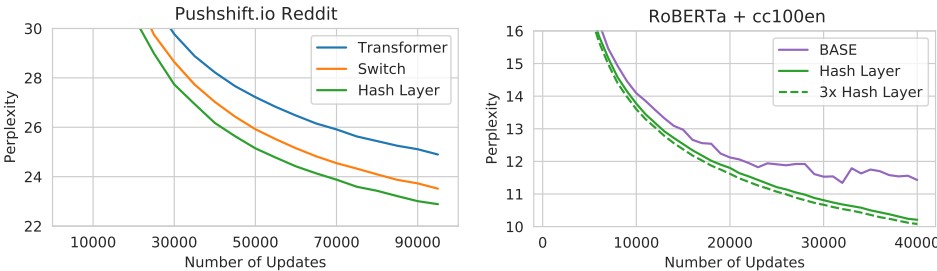

Figure 2: **Comparison of Hash Layers with other models**. (left) Validation perplexity of a baseline Transformer, Switch Transformer, and Hash Layer on the pushshift.io Reddit dataset with 128 modules. (right) Validation perplexity of BASE, Hash Layer, and a deeper Hash Layer model on the RoBERTa+cc100en dataset. All sparse models have the same number of parameters.

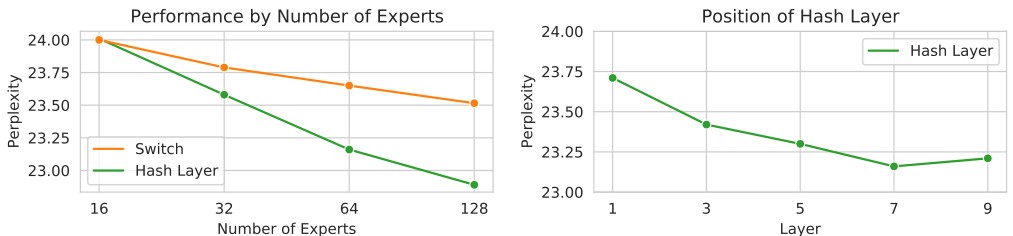

Figure 3: **Comparing Different Number of Expert Modules and Layer Position**. We compare (left) the validation perplexity wrt. the number of expert modules on the pushshift.io Reddit task for a Hash or Switch Layer on layer 7 of an 11 layer decoder in a Transformer. The baseline Transformer obtains a perplexity of 24.9. We compare the performance when adjusting the layer position of a 64 module Hash Layer on the same task (right). Placing on later layers works best.

**Dense vs. Sparse Models**  Both Hash and Switch sparse models outperform the dense Baseline (222M parameters) they are based on, as well as the Wider Transformer (755M parameters). However, the Deeper Transformer (755M parameters) outperforms the sparse models which have a similar number of parameters. However, we note that due to its dense rather than conditional compute it is slower in inference speed. We see this as a general trend: good dense models can get more power out of the same number of parameters than sparse models. However, sparse models, although more wasteful in memory, give better perplexity for the same speed (i.e, we should compare to the Baseline Transformer in this case, which has roughly the same amount of computation).

**Hash layer module size**  We conduct the same pushshift.io Reddit experiments as above, but altering the number of expert modules in both Hash and Switch. Increasing from 64 to 128 modules (1.28B parameters total) sees an even larger improvement of Hash over Switch (about 0.6 perplexity), see Table 1 (rows 6 and 7), and Figure 2 (left). Trying smaller numbers of modules, 16 and 32, and plotting all the results in Figure 3 (left) we see that for small numbers of modules Hash and Switch perform similarly, but the gap grows larger as the number of modules increases. For small numbers of modules, we hypothesize that learning to route, as Switch does, would be more important to be performant with those choices, but with larger numbers of modules many routing choices could work. Hence, Hash layers can work well in that setting, and learning to route becomes less important.

**Hash layer position**  We also experiment to find the best position layer-wise for the sparse routing to take place. In Figure 3 (right) we plot perplexity for the 64 module Hash Layer, placing on different layers of the decoder. We find that later layers perform better, but even the worst performing choice (layer 1) is still performing well compared to other baselines: as good as Switch Transformers using later layers in fact. We note that analysis of BASE Layers [10] showed a similar trend that later layers work well. Hypothesizing that conditional compute gives the ability to make fine-grained specializations, it follows that it is worth making those distinctions after more obvious features have first been extracted. We will return to this argument in later experiments.

Table 3: **Different Hash Layering Methods** on pushshift.io Reddit.

| Model | Hashing Type | Valid PPL | Test PPL |
|---|---|---|---|
| Baseline Transformer | - | 24.90 | 24.96 |
| Hash Layer 1x64 | Balanced assignment | 23.16 | 23.23 |
| Hash Layer 1x64 | Fixed random assignment | 23.22 | 23.27 |
| Hash Layer 1x64 | Token clustering (using Baseline Transformer) | 23.90 | 23.99 |
| Hash Layer 1x64 | Dispersed Hash (within token clusters) | 23.17 | 23.22 |
| Hash Layer 1x64 | Hash on position | 25.07 | 25.14 |
| Hash Layer 1x64 | Bigrams | 24.19 | 24.28 |
| Hash Layer 1x64 | Previous token | 24.16 | 24.22 |
| Hash Layer 1x64 | Future token predictions (using Transformer Baseline) | 25.02 | 25.09 |
| Hash Layer 1x64 | Future token (Oracle) | 1.97 | 1.97 |
| Hash Layer 5x16 | Same hash per layer (balance assignment) | 23.74 | 23.81 |
| Hash Layer 5x16 | Different Hash per layer | 23.21 | 23.27 |

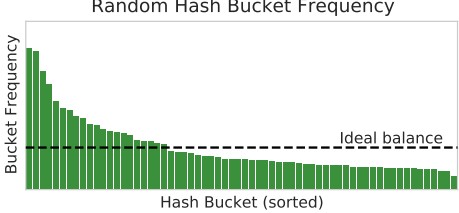 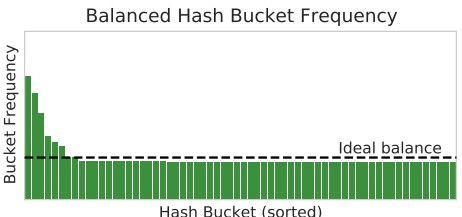

Figure 4: **Relative frequency for 64 expert modules** with Random Hash (left) and Balanced Hash (right). The Zipfian distribution makes perfect balance impossible, but Balanced Hash is closer.

**Multi-layer routing**    We evaluate placing sparse routing every other layer, 16 different modules each in Table 1 (rows 8-10). Switch and Hash perform similarly in this setting, with Switch outperforming with the optimal choice of 0.1 load balancing (23.00 vs. 23.21), and the same performance (23.19) for balancing parameter 0.01. Given the results of Figure 3 (left), the small number of modules in this case may make performance close.

**Downstream fine-tuning**    We compare several of the pushshift.io Reddit models for the goal of fine-tuning on downstream tasks. We experiment with either fine-tuning the whole model, or freezing some parts of the model during fine-tuning, as well as altering the load balancing for Switch at fine-tune time. Results are given in Appendix A. We find that the fine-tune results generally agree with the original performance on the pre-training pushshift.io Reddit task, and the order of methods is retained. Hash outperforms Switch slightly, both outperform the Baseline model, and the larger dense models perform better, as expected. Freezing parts of the model generally hurts fine-tuning, unless the part frozen is the sparse part of the model. It appears in that case just fine-tuning the dense parts of the model is sufficient for good performance. Only tuning the sparse part of the model, on the other hand, hurts performance, perhaps because the majority of the capacity of the model lies there.

### 5.3.2   Hash Function Analysis

We evaluate the different choices of hashing function detailed in subsection 3.1. The overall results are given in Table 3 on the pushshift.io Reddit dataset using a 64 module Hash Layer.

**Random and Balanced Hash Functions**    We find that fixed random assignment (row 3) and balanced assignment (row 2) perform similarly well in terms of perplexity (23.22 vs. 23.16 valid perplexity). However, balanced assignment, as its name suggests, is more balanced, see Figure 4, which may render it more efficient in terms of distributed training schemes.

**Clustering Hash Functions**    Interestingly, using cluster based hashes ("Token clustering", row 4) performs clearly worse than randomized hashes (23.90 vs. 23.22). We hypothesize that if the goal of conditional computation is to make fine distinctions, then those distinctions are more likely to appear between tokens *within* the same cluster, hence they should be in different hashes (parts of the compute graph), not the same one. We provide partial evidence for this by hashing within token clusters instead ("Dispersed Hash", row 5), which restores the performance to be similar to random

Table 4: **Comparison of Models on Wikitext-103.** We compare a baseline dense Transformer to our sparse models, which have 1 sparse layer with 16 modules (1x16). We show results with two different dictionaries, the BB [34] BPE dictionary (8008 tokens) and the standard one for the task (267,739 tokens). As these are different dictionaries, perplexities are not comparable across columns.

| Model | Configuration | Std. Dict Valid PPL | BB Dict Valid PPL |
|---|---|---|---|
| Baseline Transformer | layers=8, $d$=512, $D$=512 | 33.09 | 12.58 |
| Switch Transformer | layers=8, modules=1x16, load_bal=0.1 | 31.76 | 11.67 |
| Hash Layer | layers=8, modules=1x16 | 32.32 | 11.58 |

hashes (23.17 vs. 23.22). We note that learn-to-route methods such as Switch Transformers and BASE use simple functions of the hidden state to perform routing, which generally provide clustered expert modules [10], which could hence be a disadvantage for those methods.

**Position-based Hash Function** We conduct experiments hashing based on sequence position only. We consider this experiment as a sanity check, we did not expect choosing conditional compute based on position in the output sequence to help. Indeed, it turns out that this is no better than the dense Transformer baseline. Thus it appears that routing based on input content is much more important.

**Bigram Hash Function** Hashing based on the last two tokens (bigrams) performs worse than using only the last token (24.19 vs. 23.16). We hypothesize there are two reasons for this: (1) first, the last token is clearly the most pertinent, and bigrams add a less relevant feature; (2) this creates too many hashes, which performs less well. Subsequent experiments will help test these claims.

**Previous Token Hashing** Hashing based on the previous token is clearly worse than using the current token (24.16 vs. 23.16), and gives similar performance to using bigrams, helping confirm the first part of our above bigram hypothesis.

**Dictionary size** We perform experiments on Wikitext-103 in two settings: using the given dictionary of 267k tokens, or using the 8k dictionary we use in our pushshift.io Reddit experiments, following [34]. The results, comparing to Switch and a baseline Transformer, are given in Table 4. We find that Hash works well for the small dictionary, slightly outperforming Switch. However, on the larger dictionary, it performs *worse* than Switch. As this is the same data but just the tokenization has changed we conclude the hashing induced from the smaller dictionary is easier to learn from, helping confirm the second part of our above bigram hypothesis.

**Oracle Future Token Hashing** We evaluate hashing using the oracle next token that is to be predicted. This yields a perplexity of 1.9. Using oracle information just to choose between modules is sufficient to essentially solve a task.

**Predicted Future Token Hashing** The last result poses the question: *if we can predict the next token, and hash based on that prediction instead – will it be better than hashing on the current token?* We thus tried hashing using the Baseline Transformer to predict labels, yielding a perplexity of 25.02 – which does not actually beat the Baseline itself. It appears that the bias of the token predictions limits the ability of the sparse routing to improve.

**Multi-hashing** We evaluate the multi-hashing technique described in subsection 3.2. Results are given in Appendix A, comparing to Switch and standard hashing. Even though the same number of parameters is used in all cases, we see improvements for splitting the hash into 2, 4 or 8 different hashes compared to a single hash, with steadily improving results for both 16 or 32 modules.

### 5.3.3 Switch Transformer Analysis

**Switch load balancing** We show the performance of Switch for different values of the load balancing parameter on pushshift.io Reddit in Appendix A. Clearly the choice of parameter is important, with results varying over a 1 perplexity point range.

**Switch with Token-based Routing** Given our analysis of oracle and predicted token hashing in subsection 5.3.2, we hypothesize that the hidden representations in layers of the Transformer, being biased towards the predictions of the model, may be suboptimal for routing. We therefore experiment with a hybrid between Switch and Hash Layers: on the sparse layer, instead of using hidden state as

Table 5: **Multi-hashing experiments** on pushshift.io Reddit. When multi-hashing, the same number of parameters is used, but the FFN weights are split and indexed into multiple hashes and then concatenated together for the forward step.

| Model | Configuration | Params | Valid PPL | Test PPL |
|---|---|---|---|---|
| Switch Transformer | layers=11,modules=1x32, load_bal=0.1 | 483M | 23.79 | 23.84 |
| Hash Layer | layers=11,modules=1x32 | 483M | 23.58 | 23.65 |
| MultiHash Layer | layers=11,modules=1x32,hashes=2 | 483M | 23.48 | 23.53 |
| MultiHash Layer | layers=11,modules=1x32,hashes=4 | 483M | 23.38 | 23.45 |
| MultiHash Layer | layers=11,modules=1x32,hashes=8 | 483M | 23.28 | 23.34 |

Table 6: **Switch Transformers with Token-Based Routing** on pushshift.io Reddit. We compare standard Switch which routes based on the hidden state to token feature-routing ('Token Switch').

| Model | Configuration | Params | Valid PPL | Test PPL |
|---|---|---|---|---|
| Switch Transformer | layers=11,modules=1x64, load_bal=0.1 | 751M | 23.65 | 23.73 |
| Token Switch | layers=11,modules=1x64, load_bal=0.1 | 751M | 23.43 | 23.43 |
| Switch Transformer | layers=11,modules=1x128, load_bal=0.1 | 1.28B | 23.52 | 23.58 |
| Token Switch | layers=11,modules=1x128, load_bal=0.1 | 1.28B | 23.26 | 23.32 |

the Switch router input, we use the current token instead. To convert the token to a vector we use an extra lookup table, i.e., an extra set of learnable parameters that is the size of the dictionary. These parameters are independent of the hidden state and are only used by the router to learn the best route. Results are given in Table 6. We find this brings some small improvements to Switch for 64 and 128 modules on a single layer, affirming the usefulness of token-based routing.

### 5.3.4 Comparison to BASE Layers

We next compare to BASE Layers. Using the BASE Layer code base, we implement Hash Layers in exactly the same setup, changing only the routing method, and leaving everything else fixed. Figure 2 (right) shows results comparing Hash with BASE for 4.5B parameter models. Across the entire run, we see that Hash outperforms BASE at each training step. During early parts of training, Hash would presumably have an advantage in being able to specialize expert modules earlier, while BASE must learn membership for each of the expert modules. Later in training, BASE becomes mildly unstable presumably as expert assignments shift, while Hash performance continues to improve smoothly.

Additionally, to demonstrate Hash Layers remain performant when stacked, we trained a model with 3 Hash Layers (using random hashes), but fewer parameters per expert module so the total parameters remained constant at 4.5B (see subsection B.2). We find that using multiple Hash Layers gives a small but consistent improvement, suggesting Hash Layers will be effective at even more depth.

In addition to performance gains compared to BASE, we also find that Hash Layers are more efficient in total computation. In particular, BASE requires two all-to-all communications: the first de-correlates batches in order to make assignment balancing more stochastic, and the second routes states to their assigned expert. As Hash Layers use fixed, pre-computed assignments they avoid the de-correlation step. In practice, we find this gives an improvement of about 11% in updates-per-second. As the number of expert layers increases, this difference will become more exaggerated.

## 6 Conclusion

We have introduced a simple and efficient approach to sparse models in the Transformers-for-NLP setting based on hash layers. We showed on a variety of datasets and with analysis in various settings that this approach is highly competitive with existing methods such as Switch Transformers and BASE Layers, whilst being robust and far simpler – requiring no extra learning parameters, assignment algorithm or changes to the objective function. Given that researchers typically have only one opportunity to train very large models, this makes our approach a strong candidate for such runs. While our experiments scale up to 4.5B parameters, we do not reach the scales of large industrial works such as [8], and we hope to see future work conduct such experiments. Finally, given that our routing approach is learning free, our results perhaps suggest that none of the current approaches are routing particularly well. We thus believe learning-to-route should continue to be the study of future work, and consider our work a strong baseline for such research.

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
