# A Additional Results

Table 7: **Multi-hashing experiments** on pushshift.io Reddit. When multi-hashing, the same number of parameters is used, but the FFN weights are split and indexed into multiple hashes and then concatenated together for the forward step.

| Model | Configuration | Params | Valid | Test |
|---|---|---|---|---|
| Switch Transformer | layers=11,modules=1x16, load_bal=0.1 | 348M | 24.00 | 24.13 |
| Hash Layer | layers=11,modules=1x16 | 348M | 24.01 | 24.06 |
| MultiHash Layer | layers=11,modules=1x16,hashes=2 | 348M | 23.88 | 23.93 |
| MultiHash Layer | layers=11,modules=1x16,hashes=4 | 348M | 23.73 | 23.80 |
| MultiHash Layer | layers=11,modules=1x16,hashes=8 | 348M | 23.83 | 23.88 |
| Switch Transformer | layers=11,modules=1x32, load_bal=0.1 | 483M | 23.79 | 23.84 |
| Hash Layer | layers=11,modules=1x32 | 483M | 23.58 | 23.65 |
| MultiHash Layer | layers=11,modules=1x32,hashes=2 | 483M | 23.48 | 23.53 |
| MultiHash Layer | layers=11,modules=1x32,hashes=4 | 483M | 23.38 | 23.45 |
| MultiHash Layer | layers=11,modules=1x32,hashes=8 | 483M | 23.28 | 23.34 |

Table 8: **Fine-tuning Dense and Sparse Models in various configurations** on the BST Tasks.

| Model | Configuration | Params | BST Valid |
|---|---|---|---|
| Baseline Transformer | layers=11, $d$=1024, $D$=4096 | 222M | 14.21 |
| Wider Transformer | layers=11, $d$=2048, $D$=6144 | 755M | 12.48 |
| Deeper Transformer | layers=22, $d$=1536, $D$=4096 | 755M | 12.83 |
| Switch 1x64 | No weights frozen, load_bal=0.0 | 751M | 13.67 |
| Switch 1x64 | No weights frozen, load_bal=0.1 | 751M | 13.67 |
| Switch 1x64 | Switch weights frozen | 751M | 13.65 |
| Switch 1x64 | Router weights frozen | 751M | 13.61 |
| Switch 1x64 | All layers but last frozen | 751M | 14.42 |
| Switch 1x64 | All layers but Switch frozen | 751M | 14.37 |
| Hash 1x64 | No weights frozen | 751M | 13.45 |
| Hash 1x64 | Hash weights frozen | 751M | 13.56 |
| Hash 1x64 | All layers but last frozen | 751M | 14.29 |
| Hash 1x64 | All layers but Hash frozen | 751M | 14.12 |

Table 9: **Switch Transformer Load Balancing.** We show the perplexity with 64 modules on the pushshift.io Reddit task for different load balancing parameters. The choice of parameter is important; without balancing the model performs worse.

| Model | Load balance | Valid | Test |
|---|---|---|---|
| Baseline Transformer | - | 24.90 | 24.96 |
| Switch | 0 | 24.80 | 24.86 |
| Switch | 0.01 | 23.95 | 24.01 |
| Switch | 0.05 | 23.68 | 23.74 |
| Switch | 0.1 | 23.65 | 23.73 |
| Switch | 0.5 | 23.68 | 23.74 |

# B Hyperparameters

## B.1 Comparisons to Switch

We give here the parameters used in our standard pushshift.io Reddit and RoBERTa+cc100en setups. Other experiments with parameter changes differing from these are indicated in the main text.

| Hyperparameter | Switch | Hash Layer |
|---|---|---|
| Total parameters | 751,224,896 | 751,159,296 |
| Expert Modules per MoE layer | 64 | 64 |
| Number of MoE layers | 1 | 1 |
| FFNs per Expert Module | 1 | 1 |
| Embedding Size | 1024 | 1024 |
| FFN Size | 4096 | 4096 |
| Attention Heads | 16 | 16 |
| Number of encoder layers | 2 | 2 |
| Number of decoder layers | 11 | 11 |
| Context Length | 128 | 128 |
| Label Length | 128 | 128 |
| Batchsize | 40 | 40 |
| Gradient Accumulation | 1 | 1 |
| Maximum LR | 0.002 | 0.002 |
| Warmup | 10,000 steps | 10,000 steps |
| LR Scheduler | InvSqrt | InvSqrt |
| Maximum steps | 100,000 | 100,000 |
| Optimizer | ADAM | ADAM |
| Gradient Clip | 1.0 | 1.0 |

## B.2 Comparisons to Base

| Hyperparameter | BASE | Hash Layer | 3x Hash Layer |
|---|---|---|---|
| Shared parameters | 1,313,460,224 | 1,313,460,224 | 1,313,460,224 |
| Parameters per Expert | 100,706,304 | 100,706,304 | 33,568,768 |
| Total parameters | 4,536,061,952 | 4,536,061,952 | 4,536,061,952 |
| Expert Modules per MoE layer | 32 | 32 | 32 |
| Number of MoE layers | 1 | 1 | 3 |
| FFNs per Expert Module | 3 | 3 | 1 |
| Embedding Size | 2048 | 2048 | 2048 |
| FFN Size | 8192 | 8192 | 8192 |
| Attention Heads | 16 | 16 | 16 |
| Number of shared layers | 24 | 24 | 24 |
| Context Length | 1024 | 1024 | 1024 |
| Batchsize | 2 | 2 | 2 |
| Gradient Accumulation | 4 | 4 | 4 |
| Total tokens per update | 512k | 512k | 512k |
| Maximum LR | 7.5e-4 | 7.5e-4 | 7.5e-4 |
| Warmup | 2000 steps | 2000 steps | 2000 steps |
| LR Scheduler | Poly Decay | Poly Decay | Poly Decay |
| Maximum steps | 62,500 | 62,500 | 62,500 |
| Optimizer | ADAM | ADAM | ADAM |
| Gradient Clip | 0.1 | 0.1 | 0.1 |

Note that within the comparisons to BASE, we utilize BASE's gradient clipping method, which computed gradient norm based only on *shared* parameters to avoid additional communication across devices.

## C  Computational Resources

All experiments were run on an internal cluster. Unless otherwise marked, all experiments used 8 32GB V100 GPUs for roughly 20 hours.

Exceptions:

- Larger dense Transformer baselines and 128 module experiments used 16 V100s.
- The comparisons to BASE use 32 V100s for approximately 2 days.

## D  Societal Impact

Improvements to language modeling could have implications on a large number of surfaces across humanity. Hash Layer may also be used to train much larger models, which may have an increased impact on the environment, albeit at a fraction cost than the parameter-equivalent dense models. Hash Layer also offers a nontrivial reduction in computational resources over the prior work of BASE.

The datasets used in this work contain varied and potentially offensive text content, as they were originally procured from the Internet by third parties. Mitigating the negative effects of these efforts is an important research area, but outside the scope of this paper. We expect (but do not show) that such mitigation efforts are likely orthogonal and complementary to our own work on architecture improvements.