# OpenReview forum: "Hash Layers For Large Sparse Models"
_NeurIPS.cc/2021/Conference — NeurIPS 2021 Spotlight_

### Official Review · Reviewer_UiAT · 2021-07-15

**Rating:** 9
**Confidence:** 4

**Summary:**

**Summary And Contributions:**

This paper is interested in routing strategy in MoE Transformers in NLP where each tokens of the sequence are “routed” to a specific FFN (also called expert). The authors argue that the difficulty of learning parametrized routing strategies (especially as the number of experts grows) calls for simpler baselines. The experiments show that a simple hashing baseline (pre-defining the expert routing for each token based on some token-frequency rules) is an easy-to-implement, robust and strong baseline that is either on par or outperforming learned routing strategies. Most of the comparisons are done with the perplexity on a variety of datasets and the authors also analyze a series of different hashing strategies.



**Limitations And Societal Impact:**

The authors adequately addressed the limitations and potential negative societal impact of their work.

**Main Review:**

**Strengths:**

- The paper is well organized and written, and it’s easy to follow.
- Experiments are quite exhaustive and convincing, giving to a strong and easy to implement baseline for future research on Transformer MoE

**Weaknesses:**

- While the differences in perplexity are noteworthy, it is less convincing that there is a difference in downstream fine-tuned performance. It is not straightforward that further gains in perplexity are interesting, especially from the perspective of memorization (that the authors briefly touched upon in the appendix).
- It would be valuable to extend the comparisons to more fine-tuning setups which would bring a more useful signal from the perspective of applications (compared to perplexity), especially since it is not clear that gains in perplexity necessarily translate to gains in downstream performance (fine-tuned or zero-shot).

**Questions:**

- Training very large dense language models (more than a billion parameters) is known to be unstable (the loss can diverge unexpectedly during the training). Did you notice any differences in training stability in hashing vs learned-routing strategies?
- How would this approach be extended to multilingual setups? Any thoughts on how to scale the number of experts? The dictionary size?
- Any thoughts on the memorization abilities of the obtained models? Is it possible that improving the routing makes it easier for the model to memorize random facts and training instances?
- Is the balancing additional loss used during fine-tuning too?


**Time Spent Reviewing:**

2

---

> ### Author Response · Authors · 2021-08-09
> **Downstream performance; stability; multilinguality/vocab size; memorization; clarification on fine-tuning**
>
> Thank you for your consideration and comments.
>
> We agree the differences in performance on downstream fine-tuning are less dramatic than the original language-modeling differences. However, our primary goal in showing the fine-tuning results is that the fixed routing schemes do not come at a cost of downstream performance, and that the direction of performance agrees with the general pre-training experiments. Methods for better fine-tuning MoEs is an important research direction, but outside the scope of our work. We also note that several papers [1,2] have found positive correlation between perplexity and downstream performance, including within open-domain chit-chat.
>
> [1] https://arxiv.org/abs/2001.09977
> [2] https://arxiv.org/abs/2005.14165
>
> In our experience, training dense language models do not experience instability when using the prelayernorm Transformer architecture. In all our experiments in this paper, with both dense and sparse models, we use the prelayernorm architecture.
>
> Extending Hash layers to multilingual models is an interesting question and we do not have any empirical results on this at the moment. Based on our findings, Hash layers perform suboptimally when the dictionary becomes too big relative to the number of experts. This is particularly clear in our WikiText-103 results, which has a vocabulary of >200k. For the BPE-based dictionaries we tried (8k BPE tokens for the Reddit tasks, 52k tokens for the Roberta tasks), we find Hash layers work well within the range of expert counts we tried (8—64). This suggests there is definitely a “sweet spot” for vocabulary size.
>
> We hypothesize Hash Layers is probably slightly better at memorization than learned routing for the early stages of training. We believe this may be the case as examples may be “forgotten” as the routing distribution changes in training, due to previously-seen training examples whose expert assignment changed. As one small piece of evidence, we found parameter-equivalent BASE and Hash Layer models to have 14.57 and 14.36 training ppl respectively at 40k updates.
>
> Table 8 shows fine-tuning results for both with and without load-balancing with no weights frozen, and we find it makes no difference. For the remaining experiments in Table 8, we used both and reported the better performing model.

---

### Official Review · Reviewer_sntW · 2021-07-16

**Rating:** 7
**Confidence:** 4

**Summary:**

The authors propose using hashing layer to replace the routing mechanism in the methods of mixture-of-expert such as Switch
Transformers and BASE Layers. The main benefit of the proposed method is that no routing parameters need to learn and achieve quite good performance on the language modeling tasks. The authors also have a wide exploration of different hashing mechanisms, such as random hash, balanced hash, bigram hash, clustered hash, etc.

**Limitations And Societal Impact:**

1. Hasn't proved to be a general framework for different NLP tasks, only testing on language modeling tasks.


**Main Review:**

Overall, the proposed method is interesting and easy to use. It's promising for the community to further boost the performance of super-large models. The authors have also done a solid comparison between the switch-transformer and wider/deeper models. However, my major concern is why this method can be better than switch-transformer which theoretically can learn more knowledge to route than simple hashing. I think there are some methods to make a better hashing function, such as balanced hashing. Could these methods also apply to switch-transformer to further boost performance?

Other comments:
1. The proposed model is only tested on language modeling tasks. I am curious about the performance of the downstream tasks, such as GLUE after pertaining with cc100 corpus.
2. The hashing function has been adopted to build sparse attention in Transformer by the works of Reformer and Routing-Transformer. The authors adopt a similar method but focus on the MLP layer in SwitchTransformer. I would say this is new and interesting, but not very novel.
3. The authors have proposed many different hashing methods. However, the simplest hashing method still works almost the best. The relatively new methods don't work well. Most experiments on other datasets are required if the authors want to claim novelty on different hashing functions. Currently, it is only evaluated on Reddit which is also not a popular LM data.

Questions:
1. Are all layers using hashing layers? What are the detailed structures of using hashing layer?
2. Any challenges when training hashing layers? Is it easy or stable to start training?

**Time Spent Reviewing:**

1.5

---

> ### Author Response · Authors · 2021-08-09
> **Downstream performance, novelty of hashing, different corpora, multiple hash functions, stability**
>
> Thank you for your comments and considerations.
>
> We do report downstream performance when fine-tuning these models on the BlendedSkillTalk dataset, though we acknowledge this is also language modeling. We agree evaluating other downstream tasks, such as GLUE, is important, but not the primary concern of our paper.
>
> We agree there are related methods, such as Latent Semantic Hashing as used in Reformers, or the Hashing Trick [1], which both exist widely in the literature. Our primary focus in our paper is not the novelty of using hashing in ML or NLP; but rather, to show that very simple, fixed-routing methods can outperform recent learned-routing approaches. We do not claim novelty for any of the hashing variants we propose — rather, we use different hashing functions for analysis of how different routing choices affect performance. Taken together, we hope our work will inspire the community to focus on improvements to both learned- and fixed-routing schemes in future MoE research and compare with stronger baselines.
>
> While Reddit was our primary testbed for our experiments, we also compare against Switch transformers in the standard WikiText-103 corpus, and against BASE using the standard RoBERTa corpus.
>
> In most of our experiments, we use a single MoE layer with a number of shared, dense layers (See Figure 3 right for an analysis of placement of the layer). However, in Table 1 (modules=5x16), Table 3 (4x16), and Figure 2 right (3x Hash Layer), we used multiple Hash Layers interspersed evenly between the dense layers.
>
> We generally did not find Hash Layers to be more difficult to train or unstable than compute-equivalent dense models.
>
> [1] https://en.wikipedia.org/wiki/Feature_hashing

---

### Official Review · Reviewer_S1J2 · 2021-07-17

**Rating:** 7
**Confidence:** 5

**Summary:**

The paper studies a simple hash-based approach to route input tokens in sparsely-gated mixture-of-experts models. The proposed approach for routing is a static hash-based approach, input tokens are hashed to 1 of the several experts in the MoE layer without the need for any learnable routing network parameters  which are typically used in MoE layers. This approach is compared against Switch Transformers and BASE layers [1, 2] on a generative dialogue task on the Pushshift.io dataset, language modeling on Roberta+cc100 and Wikitext-103 datasets and on fine-tuning on the BST dialogue tasks. On all these tasks hash-based routing is either competitive with or shows slight improvements over the previous (more complex) routing approaches.

In addition to the comparison against Switch and BASE layers, the paper also conducts some analysis on:
1. Several approaches for hash-based routing, and finds that simple balanced or random hashing performs better than approaches that take into account the preceding tokens or predicted future tokens for routing.
2. The effect of the number of experts (or the amount of sparsity) on the performance of Switch layers and Hash layers, and demonstrate that learnt routing layers are more competitive at lower levels of sparsity.

References:

[1] Switch Transformers: Scaling to Trillion Parameter Models with Simple and Efficient Sparsity, Fedus et al.

[2] BASE Layers: Simplifying Training of Large, Sparse Models, Lewis et al.

**Limitations And Societal Impact:**

Yes.

**Main Review:**

Pros:

1. The paper makes an original and valuable contribution, demonstrating that a simple hashing based approach can be competitive with more computationally expensive and complex routing approaches for MoE layers.
2. Provides a strong and simple baseline for future work on routing in MoEs, while highlighting a major weakness of MoE models being studied in literature.
3. Clearly written, easy to understand and contains several interesting ablations and analysis.

Cons:
1. The paper is limited in scope to a particular type of architecture and task. The analysis focuses on MoE layers in either decoder-only LMs, or encoder-decoder models with very light-weight encoders. The increased capacity is also limited to layers with causal attention, which might affect the need for more complex routing strategies. It's not clear how well hash layers or non-learned routing strategies would perform on tasks where MoE layers have shown major improvements in recent work, like NLU tasks with span-mask pre-training [1] or multilingual MT [2,3].

Other questions/comments for authors:

1. For figure 3b, is there a corresponding plot with position vs performance for switch layers as well? Are the trends any different from hash layers?
2. Line 200: "we hypothesize ... work": Or perhaps this indicates that routing functions are easier to optimize with top-k style gating at lower levels of sparsity / fewer experts?
3. Line 205: "We find ... work well" this finding is also similar to that observed for PKM. [4]

References:

[1] Switch Transformers: Scaling to Trillion Parameter Models with Simple and Efficient Sparsity, Fedus et al.

[2] GShard: Scaling Giant Models with Conditional Computation and Automatic Sharding, Lepikhin et al.

[3] Exploring Routing Strategies for Multilingual Mixture-of-Experts Models, Kudugunta et al.

[4] Large Memory Layers with Product Keys, Lample et al.

**Time Spent Reviewing:**

2.5

---

> ### Author Response · Authors · 2021-08-09
> **Applicability to non-causal LM. Comparisons with switch/gshard**
>
> Thank you for your comments and consideration.
>
> It is not obvious to us that Hash Layers are limited to models involving causal attention. Since our most-successful hashing functions (balanced and random) simply depend on the input token at timestep T, we could also easily apply our approach in the encoder. However, for the LM modeling, even with seq2seq models (as we frequently test), it makes the most sense to place experts in the decoder. As another alternative, we could also use easily use Hash layers with either T5 or BART style seq2seq denoising losses.
>
> For Figure 3B, we did not include a direct comparison to Switch, as these were our by-far most expensive experiments (requiring 64 gpu-days to run, compared to 6 gpu-days in our other experiments). Instead, we relied on the extensive comparisons to Switch transformers reported in the original BASE paper on the same corpus, and the direct comparisons with Switch in our other experiments. Similarly, we also did not consider a comparison to Gshard (aka top-k) as BASE compared directly with Gshard.
>
> Thank you for noting the congruency with PKM. We will add this note.

---

> > ### Comment · Reviewer_S1J2 · 2021-09-11
> > **Clarification on non-causal LM and hash layers**
> >
> > Just to further clarify my point regarding applicability to non-causal layers; I didn't mean to convey that hash layers are not applicable to non-causal attention layers, but the fact that in this study the comparison is limited to causal layers. Bi-directional attention might require the need for more complex routing strategies, and the comparison between Base / Switch routing and hash layers might end up favoring more expressive routing strategies when bidirectional context is available. This is not a limitation of the technique but a limitation in the scope of the study that should be identified.
> >
> > Independent of this point, I believe the paper is a valuable contribution to the community and would maintain my current recommendation to accept.

---

### Official Review · Reviewer_eK9M · 2021-07-25

**Rating:** 7
**Confidence:** 5

**Summary:**

Paper studies the routing problem in sparsely gated mixture of experts model for text. The central contribution of the paper is experimenting the efficacy of deterministic routing strategies that are not learned over the course of training but fixated at the beginning of training by utilizing hash functions. Various hash functions are experimented on four different tasks, and compared against prior art methods (BASE layers and Switch Transformer).

- Overall well written paper with a focused contribution
- Conclusion has a more compelling and striking story which would be great to hear early on: learning free approaches for routing are quite competitive hinting that the learned routing algorithms may not be doing what we expect them to be doing.
- Paper suggests the proposed algorithm should be considered as a baseline to other methods, and given the results I cannot disagree with the proposal.

**Main Review:**

Uses a hash function on the "tokens" to decide the expert being routed throughout the network, where each expert being represented by a bucket in the hash map.

Various different types of hashing functions experimented/proposed: random, balanced, bi-gram hashing, previous-token hash or hashing based on position, clustered or dispersed hashes etc.

**Questions**
- What is the effect of vocabulary size, vocab composition to the final quality?
- If we were to switch to character or byte level vocabularies, how could we apply the proposed algorithm?
- What would be the extension you could propose if we were to extend this to vision? While you criticize the learned routing methods, they are also widely applicable to other domains. Could you please elaborate on this.
- If we were to use multiple moe layers, then all the tokens are always going to be sharing the same expert for each layer right?
 Multihash layers, --> caveat, you may end up activating the entire parameter space, will have a downside on the inference, but also each token will have the chance to touch all the experts, slowing down the training (practically making it a fully dense model if $N$ is equal to the number of experts, and each hashing method assigns a token to a separate expert).

**Typos**
- line 74: advanced -> advance

**Time Spent Reviewing:**

4 hours

---

> ### Author Response · Authors · 2021-08-09
> **Vocab size, applicability outside language modeling, multiple hash layers**
>
> Thank you for your comments and considerations.
>
> We do find, as shown in the WikiText-103 experiments, that performance of Hash Layers degrades with using larger vocabularies and a fixed number of experts. We hypothesize the pigeonhole principle becomes increasingly problematic as the vocabulary increases. Although we do not report it, we also evaluated Hash Layers for character level language modeling (hashing on the character ID instead of the token ID), also found performance degraded. This suggests that using BPE-based dictionaries, with vocabularies on the order of 10-50k, is hitting a sweet spot.
>
> Although there is a huge literature on hashing patches from images, and some of these techniques may play well with the approach in our paper, we agree that it may be difficult to find an equivalent deterministic routing function for Vision Transformers.  We provide no evidence one way or another towards such claims, and have not done these experiments.  In general though, we would advocate less for Hash Layers specifically, and more for stronger, simple baselines that provide analytical value and insight.
>
> We perform some experiments in the setting of multiple MoE layers (see Table 1 modules=5x16, Table 3 modules=5x16, and Figure 3b). Specifically, in Table 3, we compared using the same hash function on every layer where tokens were routed consistently across depth, and separate random hash functions per layer. We find using separate functions across depth performs better. In Figure 3b, we also use separate random hashes in our 3x experiment.

---

### Decision · Program_Chairs · 2021-09-27

**Decision:**

Accept (Spotlight)

**Comment:**

This work presents a surprising result that a deterministic routing strategy for sparse MoE models can yield competitive results for language tasks compared to learnable approaches. The experiments conducted are extensive and well thought. Regarding using perplexity as the only comparison metric, I suggest the authors either consider additional metrics (e.g., qualitative/human evaluation) or using multiple perplexity values, e.g., in Figure 2 or 3, to make a conclusion. The reason is a single small difference of < 1 ppl point (at the range of ppl > 20) might not be conclusive. Nevertheless, I agree with the reviewers that this paper is well written and the proposed approach serves as a simple and strong baseline. Hence, I recommend Accept.